# Adapting and Validating the COVID-19 Vaccine Hesitancy and Vaccine Conspiracy Beliefs Scales in Korea

**DOI:** 10.3390/healthcare10112274

**Published:** 2022-11-13

**Authors:** Hyesung Ock, Mihyeon Seong, Insook Kim

**Affiliations:** 1Department of Pulmonary Medicine, Samsung Changwon Hospital, Changwon 51353, Republic of Korea; 2Department of Nursing, Changshin University, Changwon 51352, Republic of Korea

**Keywords:** COVID-19, self-efficacy, vaccine hesitancy, pandemic, Republic of Korea

## Abstract

The coronavirus (COVID-19) pandemic has emphasized a need to assess the cause of vaccine hesitancy. This study verified the reliability and validity of the Korean versions of the COVID-19 vaccine hesitancy scale and vaccine conspiracy belief scale and the correlation between them. The COVID-19 vaccine hesitancy scale, Korean COVID-19 vaccine hesitancy scale, vaccine conspiracy beliefs scale, and self-efficacy scale were the study tools. Following translation into Korean, back translation into English, content validity verification, and preliminary survey, valid samples were obtained from 400 adults aged >20 years. Exploratory factor analysis revealed that “belief” and “lack of trust” accounted for 62.4% of the total variance. The model fit index of the vaccine conspiracy beliefs scale revealed that all values were in a good range. The Korean version of the COVID-19 vaccine hesitancy scale showed a positive correlation with vaccine conspiracy beliefs (r = 0.74, *p* < 0.001) and a significant negative correlation with self-efficacy (r = −0.17, *p* < 0.001). The validity and reliability of the COVID-19 vaccine hesitancy scale and vaccine conspiracy beliefs scale were verified. The Korean versions of the two scales can contribute to programs that measure and mediate various factors influencing vaccination during the COVID-19 pandemic.

## 1. Introduction

As of 2022, the novel coronavirus disease (hereinafter referred to as COVID-19) outbreak has spread worldwide, and the pandemic remains ongoing [1,2]. The COVID-19 pandemic has resulted in drastic changes in society in general, and the public is becoming increasingly anxious [3]; this is an emotional response to loss, isolation, and loneliness. In addition, negative emotions, such as fear, anxiety, anger, and uncertainty, lead to other negative attitudes [3,4]. Although vaccines are considered an important achievement in the history of medicine [3,5,6], at the same time, public concern regarding vaccine-related side effects and safety has increased [6]. In Korea, vaccines including Pfizer, Moderna, and Janssen, which were approved for commercial distribution after January 20, 2020, have started being used as vaccinations, but people are still confused due to the continued emergence of mutated viruses [7,8]. Vaccines and treatments developed as a response to COVID-19 since the early stage of its spread have failed to live up to the expectations of the general public [9]. The public are reluctant to be vaccinated owing to the rapid approval of COVID-19 vaccinations, the continued emergence of different variants of the virus [3], and the influence of medical conspiracy beliefs, scientific skepticism, and distrust in government agencies [5].

Public concern regarding vaccines has led to a refusal to get vaccinated, which increases the risk to the individual as well as the community [6]. In 2012, the World Health Organization coined the term “vaccine hesitancy”, based on research, to refer to the phenomenon of varying degrees of vaccine acceptance depending on time, location, specific vaccine, and complex circumstances [10]. Despite the availability of vaccination services, delayed acceptance or the rejection of vaccination is identified as one of the major threats to public health worldwide [10,11]. Several studies have linked the cause of reluctance to be vaccinated with anti-vaccine data available online and conspiracy theories [12,13,14,15]. Moreover, belief in conspiracy theories is related to negative attitudes toward vaccination [16], and individuals exposed to anti-vaccine conspiracy theories are influenced in terms of their willingness to be vaccinated [17]. As such, belief in a conspiracy may affect one’s health decisions [18,19], and it is important to accurately assess the cause of vaccine hesitancy and to monitor this phenomenon in the context of the COVID-19 pandemic.

Therefore, it is necessary to study the behavior of the public using a standardized, validated measurement scale to achieve the overall goal of overcoming the COVID-19 pandemic. Scale validation studies, such as those using the fear of COVID-19 scale [20] measurement tool and the mask-wearing compliance measurement, the face mask use scale [21], have been conducted in South Korea; however, there has been no research conducted on vaccine hesitancy.

A scale with verified validity and reliability that can be considered as the gold standard must be used for measurement [22]; however, there is no validated Korean version of a scale that measures the refusal of vaccines or belief in conspiracies that is available to date. Not only that, but these concepts are not even in a state of being.

In this study, the COVID-19 vaccine hesitancy scale developed by Bolatov et al. [23] and the vaccine conspiracy beliefs scale developed by Shapiro et al. [24] were used to generate preliminary data to gain a better understanding of the public’s behavior and attitudes toward the pandemic after translation/adoption into the Korean language. This is expected to be useful for research and the establishment of public health policy as it provides valuable information on the behaviors and attitudes of the public toward COVID-19.

## 2. Research Purpose

The purpose of this study is to evaluate the reliability and validity of the COVID-19 vaccine hesitancy scale developed by Bolatov et al. [23] and the vaccine conspiracy beliefs scale developed by Shapiro et al. [24] with reference to the Korean version. The procedure used for this study is outlined as follows.

First, a Korean version of the scale for measuring COVID-19 vaccine hesitancy and the vaccine conspiracy beliefs scale was translated into Korean to suit the cultural and linguistic environment of Korea.

Second, we verified the reliability and validity of the Korean version of the COVID-19 vaccine hesitancy scale when translated into Korean.

Third, the reliability and validity of the Korean version of the vaccine conspiracy beliefs scale when translated into Korean were verified.

Fourth, we verified the correlation between the Korean version of the COVID-19 vaccine hesitancy scale translated into Korean and the vaccine conspiracy beliefs scale.

## 3. Materials and Methods

### 3.1. Study Design

This study is a methodological study to verify the reliability and validity of Korean vaccine hesitancy scales for adults after translating the scales into Korean based on the translation-reverse translation procedure [25], as suggested by Chapman and Carter, and then modifying them to fit the Korean situation.

### 3.2. Study Participants

South Korea has set a global example for quarantine against COVID-19 [26]; however, there is vaccine hesitancy even among medical professionals [4], and the booster vaccination rate as of January 2022 has dropped to 49.2% nationwide, which is a significant drop from 86.8% for the first vaccine [27]. Based on these data, this study conducted a nationwide study targeting 440 Korean adults over 20 years of age who understood the purpose of this study and voluntarily agreed to participate in the study by responding to participation emails sent at random. The number of participants was determined based on the rationale of Hinkin [28], who stated that 150 to 200 participants are required for exploratory factor analysis, and 130 to 300 participants are required for confirmatory factor analysis. In this study, 200 subjects were randomly assigned and selected for exploratory factor analysis and 200 subjects were selected for confirmatory factor analysis, and a dropout rate of about 20% for each group was taken into account so that a total of 440 subjects (EFA:220, CFA:220) was the target number selected. Responses from 439 subjects (EFA:220, CFA:219) were recovered and used as final data, excluding the first copy with insincere responses.

### 3.3. Study Tools

#### 3.3.1. The COVID-19 Vaccine Hesitancy Scale and Korean COVID-19 Vaccine Hesitancy Scale

The COVID-19 vaccine hesitancy scale was developed by Bolatov et al. [23] to confirm the major obstacles to vaccine acceptance among medical students and was used after obtaining permission from the first author. The scale consists of 12 questions in total, two questions for each of the six sub-items including “efficacy and stability,” “risk awareness,” “conspiracy belief,” “lack of knowledge and information,” “trust,” and “norms related to social and religious beliefs and family attitude.” This study used the COVID-19 vaccine hesitancy scale after translation and back translation to reflect the circumstances in South Korea. This scale uses a 5-point Likert scale that rates each question as “not at all” (1 point), “almost never” (2 points), “moderately” (3 points), “sometimes” (4 points), and “definitely” (5 points), where a high score indicates a high level of hesitancy toward the COVID-19 vaccine. At the time of development, Cronbach’s α value for the reliability of the COVID-19 vaccine hesitancy scale was 0.83.

#### 3.3.2. Vaccine Conspiracy Beliefs Scale

The vaccine conspiracy beliefs scale was developed by Shapiro et al. [24] to evaluate the impact of vaccine conspiracy beliefs on human papillomavirus vaccine acceptance and was used with the permission of the first author. The scale consists of seven items with one dimension. In this study, the vaccine conspiracy beliefs scale was translated and back translated to reflect the circumstances in South Korea. Each question was rated on a 7-point Likert scale that rates each question as “strongly disagree” (1 point), “disagree” (2 points), “somewhat disagree” (3 points), “moderately agree” (4 points), “somewhat agree” (5 points), “agree” (6 points), and “strongly agree” (7 points), where a higher score indicates a stronger belief in conspiracy theories. At the time of development, Cronbach’s α value for the reliability of the vaccine conspiracy beliefs scale was 0.94.

#### 3.3.3. Self-Efficacy Scale

Self-efficacy is the belief that an individual can successfully perform the behavior required to produce a certain result [29]. In this study, the self-efficacy scale developed by Sherer et al. [30] and modified and supplemented by Jeong [31] was used with the permission of the original author. The scale consists of 17 questions in total, and each question is rated on a 5-point Likert scale as “not at all” (1 point), “almost never” (2 points), “moderately” (3 points), “sometimes” (4 points), and “definitely” (5 points), where a higher score indicates higher self-efficacy. In the study by Jeong [31], Cronbach’s α value for the reliability of the self-efficacy scale was 0.94.

### 3.4. Study Procedure

#### 3.4.1. Translation–Back Translation

The scales were translated after obtaining approval from the scale developers, Bolatov et al. [23] and Shapiro et al. [24]. The original text was translated using a four-step procedure in accordance with the translation–back translation process [25], and the validity of the translation procedure was verified by re-translating the first translation back into the original language, comparing it with the original, and revising the items with differences.

The scale translation process is as follows. In the first step, a bilingual person with at least 10 years of work experience translated the original text into the Korean language. In the second step, the investigator and two professors of nursing compared the translation with the original copy to assess whether the translation was suitable for the circumstances in South Korea and whether any revisions were required for accuracy. In the third step, a doctor of nursing who is fluent in English and Korean, one professor working at a nursing college with dual citizenship, and one general adult who is fluent in dual Chinese read the translated version of the contents without seeing the original text in English. In order to increase understanding and clarity, a reverse translation of the translated version was performed again into English. Finally, the researcher and translator compared the reverse-translated tool with the original English tool and completed the final translation by verifying that there was no difference in meaning between each item.

#### 3.4.2. Verifying Content Validity

Content validity for the scale that was translated and back translated in this study was verified by eight individuals, which included three men, three women, one professor of nursing, and one professor of medicine. The content validity calculation method is based on the 4-point scale suggested by Lynn [32] that rates each question as “not relevant” (1 point), “relevance cannot be determined without item correction” (2 points), “relevant but item correction is required” (3 points), and “very relevant” (4 points). The content validity index was calculated by calculating the percentage of experts out of the total number of experts who rated each question as 3 or 4 points. All scale items scored ≥0.80, and all items were selected without deletion.

#### 3.4.3. Preliminary Survey

In order to clarify the contents of the Korean version of the COVID-19 vaccine hesitation measurement tool and the vaccine conspiracy belief measurement tool, a preliminary survey was conducted to determine the appropriateness of the questions for 40 adults, 10% of the sampled population. In the case of the preliminary survey, the average age was 52.62 ± 12.21, and the preliminary survey conducted in this study was verified by obtaining Cronbach alpha, which means internal consistency of items. The value of Cronbach alpha was 0.91. In addition, the translated questionnaire was finally used after confirming that there were no items that were difficult to answer or that could not be understood. Because there was no change in the questionnaire, preliminary survey personnel were randomly assigned and included in this survey.

### 3.5. Data Collection Method and Ethical Considerations

In this study, the survey was conducted nationwide among adults to verify the reliability and validity of the Korean COVID-19 vaccine hesitancy scale and vaccine conspiracy beliefs scale after obtaining approval (IRB NO. SCMC2022-03-009) from the Institutional Review Board of the Sungkyunkwan University Hospital. Data collection was conducted remotely, in consideration of the pandemic, between March and April 2022 among adults who were South Korean nationals aged ≥20 years. The survey was conducted through an online survey vendor, and 440 valid samples were obtained to ensure that the samples were evenly distributed across all age groups. Before data collection, an information sheet of the objectives and procedures of the study was published by the investigator to ensure that participants could read it, and the survey was conducted among those who consented after having the study objectives, details of the survey, confidentiality, right to withdraw from participation, and storage and processing of data explained to them. At the end of the survey, the participants were reassured that the data would not be used for purposes other than this study and would be handled confidentially and that they could withdraw at any time. After the collection was completed, the survey data were stored on an encrypted hard disk, and access was prohibited except for the investigator. The survey took approximately 10 min to complete.

### 3.6. Data Analysis Method

The collected data were analyzed using IBM SPSS Amos 22 (IBM Corp., Armonk, NY, USA). The details of the analysis method are as follows. The general characteristics of the participants were analyzed as descriptive statistics. The content validity of the Korean versions of the COVID-19 vaccine hesitancy scale and vaccine conspiracy beliefs scale was confirmed through the content validity index analysis, and the reliability of the Korean versions of the COVID-19 vaccine hesitancy scale and vaccine conspiracy beliefs scale was confirmed by using Cronbach’s α. In addition, construct validity verification for the items in each subcategory was conducted through confirmatory factor analysis provided by the structural equation model. For the structural validity of the Korean versions of the COVID-19 vaccine hesitancy scale and vaccine conspiracy beliefs scale, the chi-square (χ^2^), minimum discrepancy function divided by degrees of freedom (CMIN/DF), root mean square error of approximation (RMSEA), root mean square residual, standardized root mean square residual (SRMR), goodness of fit index (GFI), incremental fit index, Tucker–Lewis index, and comparative fit index (CFI) were calculated. The convergent validity of the Korean versions of the COVID-19 vaccine hesitancy scale and vaccine conspiracy beliefs scale was verified using the standardized factor loading, critical ratio, construct reliability (CR), and average variance extracted (AVE). Discriminant validity was verified using correlation coefficients and AVE values. Criterion validity was verified using Pearson’s correlation to confirm the correlation of the scales.

## 4. Results

### 4.1. General Characteristics of the Participants

To assess the general characteristics of the participants, 220 participants were used for the item and exploratory factor analyses, and 219 were used for the confirmatory factor analysis to evaluate the reliability and validity of vaccine hesitancy and vaccine conspiracy beliefs. There was no statistically significant difference in the general characteristics between the two groups of participants. The details of the general characteristics are shown in Table 1.

### 4.2. Verifying Scale Validity

#### 4.2.1. Analysis of the Exploratory Factors

During the course of item analysis, the correlation coefficient between the items and total items was calculated, and the criterion >0.30 [33] was applied. No items had less than the minimum criterion of 0.30 for the correlation coefficient between each item and the total items for the 12 preliminary items in the vaccine hesitancy scale and seven preliminary items in the vaccine conspiracy beliefs scale (Table 2). As a result of the sample suitability analysis, which determines the fit of factor analysis for the 12 items of the vaccine hesitancy scale, the Kaiser–Meyer–Olkin test (KMO) value was 0.86; Bartlett’s sphericity verification value was χ^2^ = 1149.04 (*p* < 0.001). For the seven item vaccine conspiracy beliefs scale, the KMO value was 0.94; Bartlett’s sphericity verification value was χ^2^ = 1532.17 (*p* < 0.001), which satisfied the conditions of the exploratory factor analysis. For the items in the vaccine hesitancy scale, two factors with an Eigen value of ≥1.0 were extracted through the exploratory factor analysis, which accounted for 62.4% of the total variance. All 12 items had a factor loading of >0.40; however, item 10 was removed as it appeared to be >0.50 in both factors. The two derived factors were named factors with a concept that could imply the contents of each factor’s item; factor one was “belief,” and factor two was “lack of trust,” using the factor names of the original scale. Vaccine conspiracy belief was extracted as one factor, and all factor loadings were >0.40; all seven items were selected as they accounted for 81.2% of the total variance.

#### 4.2.2. Confirmatory Factor Analysis

First, the fit index was evaluated before the fit test of the factor model. Overall, the level of fit did not meet the standard, so the model needed to be modified. To increase the fit of the model, a path was added using modification indices [MI], and squared multiple correlation was used to check the explanatory power of endogenous variables (SMC) values that were used. Consequently, items one, three, four, and five were removed as they had a value of 0.40, which was below the criterion. Subsequently, as a result of evaluating the second model fit index of the vaccine hesitancy scale, all values were in a good range (χ^2^ = 17.48 [*p* < 0.001], CMIN/DF = 2.19, SRMR = 0.05, RMSEA = 0.07, NFI = 0.76, and CFI = 0.98) [34] (Table 3). In the standardized factor loading in the secondary confirmatory factor analysis, items must be at least 0.50 to be considered desirable; the standardized factor load was 0.60 to 0.88 thus satisfying the criteria in all items. In addition, when the model fit index of the vaccine conspiracy beliefs scale was evaluated, all values were in a good range (χ^2^ = 29.38 [*p* = 0.002], CMIN/DF = 2.67, SRMR = 0.02, RMSEA = 0.09, NFI = 0.98, and CFI = 0.99) [34]. The factor load was in the range of 0.72 to 0.92; thus, the validity of the single structural factor was confirmed as having met the standard value. The AVE values of the COVID-19 vaccine hesitancy scale were 0.55 and 0.53, which satisfied the criterion of ≥0.50, and the CR values were 0.78 and 0.81 for factor one and two, respectively; for the vaccine conspiracy beliefs scale, the AVE value was 0.69 and the CR value was 0.94, confirming convergent validity [34] (Table 4).

To verify discriminant validity, the root value of the AVE and correlation coefficient (φ) of the COVID-19 vaccine hesitancy scale were confirmed. Discriminant validity was also confirmed as the correlation coefficient between factor one and factor two with r = 0.17 (*p* < 0.001), indicating that the AVE value was larger than the square (R = 0.03) value of the correlation number of the two factors [34].

In addition, the correlation between the vaccine conspiracy beliefs scale and self-efficacy was analyzed to confirm the criterion validity of the Korean version of the COVID-19 vaccine hesitancy scale. It showed a positive correlation with vaccine conspiracy beliefs (r = 0.74, *p* < 0.001), and a significant negative correlation with self-efficacy (r = −0.17, *p* < 0.001). However, the correlation with self-efficacy was not significant in the Korean version of the vaccine conspiracy beliefs scale.

#### 4.2.3. Verifying Reliability

To verify the internal consistency between the Korean COVID-19 vaccine hesitancy scale, which consists of two factors including belief (three items) and lack of trust (four items), and the Korean version of the vaccine conspiracy beliefs scale (seven items), which has only one factor, the Cronbach’s α value was calculated, and the overall reliability was found to be 0.79 and 0.94, respectively. The Cronbach’s α value for belief, which is a subfactor of the Korean version of the COVID-19 vaccine hesitancy scale, was 0.77, and the Cronbach’s α value for lack of trust was 0.81.

## 5. Discussion

This study was conducted to evaluate the appropriateness of the Korean version of the COVID-19 vaccine hesitancy scale and the Korean version of the vaccine conspiracy beliefs scale. As it is more appropriate to perform a confirmatory factor analysis when a scale whose validity has been confirmed is translated into another language [35,36], confirmatory factor analysis was also conducted in this study. Since a confirmatory factor analysis was performed based on the factors and items analyzed in the first phase by selecting different participants, cross-validation was verified in the factor analysis [37]. 

In verifying the reliability of the scales in this study, the internal consistency of a scale is considered reliable when the Cronbach’s α value is ≥0.70, and the Cronbach’s α value was 0.79 for the Korean COVID-19 vaccine hesitancy scale and 0.81 for the Korean vaccine conspiracy beliefs scale [37]; as the internal consistency of the two scales was high, reliability was confirmed. This study is meaningful in that it provides the primary basis for the reliability and validity of the two scales through item analysis, content validity, construct validity, reliability, and others for the scaling assumptions. 

To increase the fit of the Korean version of the COVID-19 vaccine hesitancy scale model, model two was constructed by checking the SMC value and deleting questions one, three, four, and five that had values of ≤0.40. The suitability indicators were generally satisfied. The deleted items were ‘I am concerned about the possible side effects after being vaccinated against the corona virus’, ‘If I have already had the corona virus, I do not need to get the corona vaccine’, ‘I don’t need the corona vaccine because I’m confident that my immune system will fight the coronavirus well’ and ‘I don’t believe the coronavirus is dangerous.’ However, the reason for their deletion from this study is that Koreans are already aware of the risk of COVID-19 as of 2022 due to the high mortality rate and continued infection [38] with more than 5.9 million deaths, as well as Korea’s disaster message notifications, and announcements of evidence of the side effects of vaccines. It is inferred that the continued information updates and broadcast publicity have not directly caused hesitation about the COVID-19 vaccination. Considering this together with the original tool, it is judged that the reasons why Koreans hesitate to vaccinate are more important than factors such as ‘effectiveness and stability’ and ‘risk perception’, and relate to factors such as their beliefs, living environment, and trust. Therefore, when using this tool in future studies on vaccines, it is necessary to reflect the socio-cultural context and perceptions of Koreans. In addition, because this item was selected in the original scale and in countries other than Korea, a repeated study is proposed.

As a result of verifying the convergence and discriminant validity of the Korean version of the COVID-19 vaccine hesitancy scale, it was determined that the contents of the items reflected the characteristics of each subfactor well because the convergence and discriminant validity of three items for “belief” and four items for “lack of trust” were confirmed. It was determined that the final seven items of the Korean version of the COVID-19 vaccine hesitancy scale in this study have structural validity. In addition, the seven items of the Korean version of the vaccine conspiracy beliefs scale, which verified the reliability and validity together to be used as the standard scale of the Korean version of the COVID-19 vaccine hesitancy scale, also demonstrated compositional validity. Moreover, the two scales were significantly correlated, indicating that criterion validity was confirmed. In addition, the Korean version of the COVID-19 vaccine hesitancy scale had a significant correlation in the criterion validity with self-efficacy based on previous studies. However, the Korean version of the vaccine conspiracy beliefs scale had no significant correlation with self-efficacy, which supports the opinion of Kroke and Ruthig [39] that the existence of conspiracy beliefs would have little or no effect on self-efficacy by eliminating potential disease prevention measures. Based on these results, it can be said that conspiracy beliefs are related to negative attitudes toward vaccination [13], and individuals with conspiracy beliefs against vaccination are affected by vaccination intentions [14]. In addition, a range of 0.40 to 0.80 for the correlation coefficient is generally recommended for the criterion validity test [40], and caution is required when interpreting the findings as the correlation coefficient may be measuring a phenomenon that is different from the external criterion that the scale is trying to measure because the Korean versions of the COVID-19 vaccine hesitancy scale and vaccine conspiracy beliefs scale showed high correlation with each other, although the correlation with self-efficacy was low. Therefore, as the gold standard scale with confirmed reliability and validity was not used in this study, a repeat study should be conducted in the future.

This study is a starting point for research on vaccination intentions and behaviors of individuals in the pandemic era in that it simultaneously developed the COVID-19 vaccine hesitancy scale and vaccine conspiracy beliefs scale, which have been verified overseas for reliability and validity, in South Korea and the validity was confirmed among adults in South Korea. As the first of its kind in South Korea, this study has significance as the validity and reliability of the Korean versions of the COVID-19 vaccine hesitancy and vaccine conspiracy beliefs scales were verified. In addition, these study results may be used as basic data for revitalizing clinical practice, community health management, intervention, and research in the future. However, because this study was conducted using an internet-based survey method during the development and validity evaluation process, differences in results and selection bias problems cannot be excluded when factors such as academic background and health status are included [41,42]. In addition, according to previous studies, it is reported that psychological factors such as environment, values, social beliefs, and guilt can affect beliefs, so further research is needed. According to Uscinski et al. [43], the belief in conspiracy theories is driven by strongly maintained social and political identities, and because ties are difficult to break, these factors are used to increase the sensitivity of people with collectivist cultural tendencies [44].

## 6. Conclusions

This study translated the COVID-19 vaccine hesitancy scale and vaccine conspiracy beliefs scale into the Korean language and evaluated their applicability among South Korean adults by verifying the validity and reliability of both scales amid the ongoing pandemic in 2022 and newly emerging infectious diseases.

The original COVID-19 vaccine hesitancy scale consists of six factors and twelve items, whereas the Korean COVID-19 vaccine hesitancy scale had two factors and eleven items; the final version had seven items after excluding four items that had a lower SMC than the criterion. In addition, the Korean vaccine conspiracy beliefs scale had seven single-factor items in the final version. The suitability of the scales was verified by checking the compositional validity of the Korean versions of the COVID-19 vaccine hesitancy scale and vaccine conspiracy beliefs scale. Therefore, the two scales are expected to contribute to the development of programs that are capable of measuring and mediating the factors influencing vaccination during the COVID-19 pandemic.

## Figures and Tables

**Table 1 healthcare-10-02274-t001:** Demographics of the participants.

Characteristics	Categories	Data Set A (n = 220) N (%) or M ± SD	Data Set B (n = 219) N (%) or M ± SD
Age (y)		40.60 ± 13.23	
Sex	Male	79 (35.9)	133 (60.7)
	Female	141 (64.1)	86 (39.3)
Marital Status	Unmarried	111 (50.5)	61 (27.9)
	Married	98 (44.5)	144 (65.8)
	Divorced	10 (4.5)	9 (4.1)
	Others	1 (0.5)	5 (2.3)
Economic Level	High	9 (4.1)	10 (4.6)
	Middle	143 (65.0)	137 (62.6)
	Low	68 (30.9)	72 (32.9)
Religion	Protestant	46 (20.9)	49 (22.4)
	Catholic	24 (10.9)	21 (9.6)
	Buddhist	24 (10.9)	32 (14.6)
	Others	1 (0.5)	1 (0.5)
	None	125 (56.8)	116 (53.0)

**Table 2 healthcare-10-02274-t002:** Corrected item-total correlation and factor loadings in exploratory factor analysis using data set A.

Scale	Item	Corrected Item-Total Correlation	Factor Loading	
KCVH-S			Factor 1	Factor 2
Factor 1	12	0.68	0.88	
	11	0.61	0.88	
	7	0.70	0.78	
	5	0.63	0.69	
	4	0.78	0.66	
Factor 2	9	0.64		0.83
	8	0.68		0.80
	2	0.73		0.69
	1	0.41		0.67
	6	0.71		0.64
	3	0.64		0.50
KVCB-S				
	1	0.85	0.94	
	2	0.87	0.92	
	3	0.88	0.90	
	4	0.90	0.88	
	5	0.93	0.87	
	6	0.92	0.85	
	7	0.84	0.84	

KCVH-S = Korean COVID-19 vaccine hesitancy scale; KVCB-S = korean vaccine conspiracy beliefs scale.

**Table 3 healthcare-10-02274-t003:** Analysis of construction validity.

Scale	Fitness	χ^2^ (*p*)	DF	CMIN/DF	SRMR	RMSEA	NFI	CFI
	Index							
KCVH-S	Model 1	235.95 (*p* < 0.001)	43	5.49	0.08	0.14	0.76	0.80
	Model 2	17.48 (*p* < 0.001)	8	2.19	0.05	0.07	0.97	0.98
KVCB-S	Model 1	29.38 (*p* = 0.002)	11	2.67	0.02	0.09	0.98	0.99

KCVH-S = Korean COVID-19 vaccine hesitancy scale; KVCB-S = Korean vaccine conspiracy beliefs scale; DF = degrees of freedom; CMIN/DF= chi-square (χ^2^), minimum discrepancy function divided by degrees of freedom; GFI = goodness of fit index; SRMR = standardized root means square residual; RMSEA = root mean square error of approximation; NFI = normed fit index; CFI = comparative fit index.

**Table 4 healthcare-10-02274-t004:** Analysis of convergent validity of items.

Scale	Factors	Items	SE (β)	NSE	SE	CR	*p*	AVE	CR
KCVH-S	Factor 1	Item 12	0.70	1.00				0.55	0.78
		Item 11	0.88	1.01	0.12	8.11	<0.001		
		Item 7	0.61	1.02	0.12	8.31	<0.001		
	Factor 2	Item 9	0.80	1.00				0.53	0.81
		Item 8	0.81	0.95	0.09	10.88	<0.001		
		Item 2	0.60	0.73	0.09	8.56	<0.001		
		Item 6	66	0.76	08	9.50	<0.001		
KVCB-S	Factor 1	Item 1	0.74	1.00					
		Item 2	0.72	1.02	0.08	12.15	<0.001	0.69	0.94
		Item 3	0.86	1.19	0.09	13.22			
		Item 4	0.90	1.20	0.09	14.00			
		Item 5	0.92	1.23	0.09	14.42			

KCVH-S = Korean COVID-19 vaccine hesitancy scale; KVCB-S = Korean vaccine conspiracy beliefs scale; SE = standardized estimate; NSE = non-standardized estimate; CR = construct reliability; AVE= average variance extracted.

## Data Availability

No new data were created or analyzed in this study. Data sharing is not applicable to this article.

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
