# Peer review of "Adapting and Validating the COVID-19 Vaccine Hesitancy and Vaccine Conspiracy Beliefs Scales in Korea"

_healthcare, 2022, doi:10.3390/healthcare10112274_

Round 1

Reviewer 1 Report

First, I congratulate the authors on their successful attempt to translate a well-known survey for their study population and validate it using statistical methods. This is indeed a tough job, but the authors did it diligently. 

There are a few suggestions:

Line 30: Authors have mentioned only three vaccine manufacturers. Several of them, or in early 2022, will be manufacturing vaccines and distributing them worldwide. It would be prudent to include all or add something like Pfizer, Modera, Janssen, and several others that have been rolled out. 

Line 97: It was interesting that the authors targeted the population >20 years old. It would be helpful if the authors could explain their reasons. Generally, for any adult population, usually, the research studies look into an adult population above 18 years old. 

Line 102-105: Authors mentioned that their study recruited 440 adults but found to have a dropout rate of 20%. But then the sentence ends with 439 participants being included in the data. Please clarify this sentence. If needed, please break it down to bring more clarity. 

Line 102-105: The data analysis shows 439 participants. The authors assigned 200 participants to the exploratory factor analysis group and 200 to the confirmatory factor analysis group. So, where are the remaining 39 participants? 

Line 216-219: This clarifies the proper allotment of those 239 participants. 

Line 155: "Bilingual office worker"- please clarify if this office worker has any special training in either of those languages. Why was this office worker chosen over the other bilingual office workers?

Line 315-322 & 389: It was interesting to see how the authors deleted four questions from their survey scale. All these four questions are pertinent to human behavior, and even after some information is available, they are still important. The Assistant Secretary of Planning and Evaluation office used the Household Pulse Survey (https://www2.census.gov/programs-surveys/demo/technical-documentation/hhp/Phase_36_Household_Pulse_Survey_ENGLISH.pdf ) sponsored by the US Census Bureau which was cited on the Center for Disease Control and Prevention website for the COVID-19 vaccine hesitancy data (https://data.cdc.gov/stories/s/Vaccine-Hesitancy-for-COVID-19/cnd2-a6zw/). This survey has those relevant questions to capture the hesitance accurately. However, the authors did mention in line 389 that there was no concept of vaccine hesitancy in South Korea yet. Without evidence, it would be hard to believe that a particular population is not hesitant to COVID-19 vaccine administration when this phenomenon was seen worldwide, including in high-income economies. Even though the authors believe it, it would be prudent to provide evidence to support their statement in line 389. 

Author Response

Thanks for the good comments.
The quality of my research seems to have improved thanks to you. It has been revised to fully reflect your opinion.

Line 30: Authors have mentioned only three vaccine manufacturers. Several of them, or in early 2022, will be manufacturing vaccines and distributing them worldwide. It would be prudent to include all or add something like Pfizer, Moderna, Janssen, and several others that have been rolled out. 

→ The authors have revised it to reflect the current situation in Korea. Please see the red text on page 1.

Line 97: It was interesting that the authors targeted the population >20 years old. It would be helpful if the authors could explain their reasons. Generally, for any adult population, usually, the research studies look into an adult population above 18 years old. 

→ The authors have revised it to reflect the current situation in Korea. Please see the red text on page 3.

Line 102-105: Authors mentioned that their study recruited 440 adults but found to have a dropout rate of 20%. But then the sentence ends with 439 participants being included in the data. Please clarify this sentence. If needed, please break it down to bring more clarity. 

→ The authors added and modify. Please see the red text on page 3.

Line 102-105: The data analysis shows 439 participants. The authors assigned 200 participants to the exploratory factor analysis group and 200 to the confirmatory factor analysis group. So, where are the remaining 39 participants? 

→ The authors added and modify. Please see the red text on page 3.

Line 216-219: This clarifies the proper allotment of those 239 participants. 

→  Described as a modification previously.

Line 155: "Bilingual office worker"- please clarify if this office worker has any special training in either of those languages. Why was this office worker chosen over the other bilingual office workers?

→ The authors added and modify. Please see the red text on page 4.

Line 315-322 & 389: It was interesting to see how the authors deleted four questions from their survey scale. All these four questions are pertinent to human behavior, and even after some information is available, they are still important. The Assistant Secretary of Planning and Evaluation office used the Household Pulse Survey (https://www2.census.gov/programs-surveys/demo/technical-documentation/hhp/Phase_36_Household_Pulse_Survey_ENGLISH.pdf ) sponsored by the US Census Bureau which was cited on the Center for Disease Control and Prevention website for the COVID-19 vaccine hesitancy data (https://data.cdc.gov/stories/s/Vaccine-Hesitancy-for-COVID-19/cnd2-a6zw/). This survey has those relevant questions to capture the hesitance accurately. However, the authors did mention in line 389 that there was no concept of vaccine hesitancy in South Korea yet. Without evidence, it would be hard to believe that a particular population is not hesitant to COVID-19 vaccine administration when this phenomenon was seen worldwide, including in high-income economies. Even though the authors believe it, it would be prudent to provide evidence to support their statement in line 389. 

→ The authors added and modify. Please see the red text on page 7-8.

Reviewer 2 Report

Regarding the article titled: “A study on the reliability and validity of the Korean versions of 2 the COVID-19 Vaccine Hesitancy and Vaccine Conspiracy Beliefs Scales”

This study aimed to evaluate the reliability and validity of the COVID-19 Vaccine Hesitancy Scale and Vaccine Conspiracy Beliefs Scale developed

-          This study answered the research question and it has some novelty in analysis and interpretations.

Line 90: Can authors elaborate more on the study design, e.g retrospective study, etc

Line 96-105: 400 persons participated in the study; nonetheless, the author later discusses 439 participants. Could you please clarify the size of the study's sample?

Line 183: What is meant by P online survey vendor?

Can authors add a description for exploratory factor analysis under data analysis?

Was the preliminary survey's sample size of 40 people included in the main study and again how was the data collected, face-to-face or online also.

Line 216-217: Can the authors be consistent in terms of the sample size.

There is no need to split the data into two groups at this time because the participants (439) all share the same strata/cluster characteristics, so table 1 can simply present the overall characteristics of the participants.

How were the particpants selected for training (EFA) and validation (CFA).

Present the numbers as 0.17, not .17, in the tables and throughout the entire document.

Can the authors under discussion explain why they believe that the items that were eliminated from the COVID-19 Vaccine Hesitancy Scale are not essential to Koreans despite the fact that earlier researchers found them to be so?

Line 308: Rather than stating definitions, discuss the findings of the investigation

Author Response

Thanks for the good comments.
The quality of my research seems to have improved thanks to you. It has been revised to fully reflect your opinion.

Regarding the article titled: “A study on the reliability and validity of the Korean versions of 2 the COVID-19 Vaccine Hesitancy and Vaccine Conspiracy Beliefs Scales”

This study aimed to evaluate the reliability and validity of the COVID-19 Vaccine Hesitancy Scale and Vaccine Conspiracy Beliefs Scale developed

-          This study answered the research question and it has some novelty in analysis and interpretations.

Line 90: Can authors elaborate more on the study design, e.g retrospective study, etc

→ The authors modified. please see the red text on page 2.

Line 96-105: 400 persons participated in the study; nonetheless, the author later discusses 439 participants. Could you please clarify the size of the study's sample?

Line 183: What is meant by P online survey vendor?

Can authors add a description for exploratory factor analysis under data analysis?

Was the preliminary survey's sample size of 40 people included in the main study and again how was the data collected, face-to-face or online also.

→ The authors modified. please see the red text on page 4.

Line 216-217: Can the authors be consistent in terms of the sample size.

→  The authors added and modify. Please see the red text on page 3.

There is no need to split the data into two groups at this time because the participants (439) all share the same strata/cluster characteristics, so table 1 can simply present the overall characteristics of the participants.

How were the particpants selected for training (EFA) and validation (CFA).

→  The authors added and modify. Please see the red text on page 3.

Present the numbers as 0.17, not .17, in the tables and throughout the entire document.

→  The authors added and modify. Please see the red text on page 6-7.

Can the authors under discussion explain why they believe that the items that were eliminated from the COVID-19 Vaccine Hesitancy Scale are not essential to Koreans despite the fact that earlier researchers found them to be so?

Line 308: Rather than stating definitions, discuss the findings of the investigation

→ The authors added and modify. Please see the red text on page 8-9.

Reviewer 3 Report

Thank you for the opportunity to review the manuscript titled “A study on the reliability and validity of the Korean versions of the COVID-19 Vaccine Hesitancy and Vaccine Conspiracy Beliefs Scales”. I believe the topic is engaging and addresses a relevant issue about the investigation of conspiracy beliefs associated to covid-19 vaccination. I found the manuscript well written and understood the authors’ intent to provide such a tool, since it may have relevant practical implications.

I have minor concerns regarding the current version of the manuscript. I propose a few minor revisions and suggestions that might ameliorate their work.

- I believe that the introductory section of the manuscript, as well as the conclusions, may be improved by integrating recent literature that has highlighted psychosocial variables that may affect the endorsement of conspiracy beliefs related to covid-19 diseases and vaccinations:

1.          Douglas, K.M. COVID-19 conspiracy theories. Group Processes Intergroup Relat. 2021, 24, 270–275. https://doi.org/10.1177/1368430220982068.

2.          Pellegrini, V.; Giacomantonio, M.; De Cristofaro, V.; Salvati, M.; Brasini, M.; Carlo, E.; Mancini, F.; Leone, L. Is COVID-19 a natural event? COVID-19 pandemic and conspiracy beliefs. Personal. Individ. Differ. 2021, 181, 111011. https://doi.org/10.1016/j.paid.2021.111011.

3.          Giacomantonio, M., Pellegrini, V., De Cristofaro, V., Brasini, M., & Mancini, F. (2022). Expectations about the “Natural Order of Things” and Conspiracy Beliefs about COVID-19. International Journal of Environmental Research and Public Health19(15), 9499. Doi: 10.3390/ijerph19159499 

- Please, avoid reporting the GFI as goodness-of-fit indicators since is deprecated.

-Provide more justification on the low fit indexes of KCVH-S Model 1 and on how Authors deal with them.

- It should be clarified how the convergent and divergent validity of the measure has been tested. Usually, these types of validity are tested through the examination of convergent and divergent associations between the measure of interest and other constructs of theoretical relevance that are potentially related to it. In this paper, this does not appear to have been done. Rather, an analysis of the usefulness of having multiple dimensions in a single measuring instrument seems to have been conducted. Therefore, I would suggest clarifying this aspect and to call the related analyses in an appropriate way.

Author Response

Thanks for the good comments.
The quality of my research seems to have improved thanks to you. It has been revised to fully reflect your opinion.

Thank you for the opportunity to review the manuscript titled “A study on the reliability and validity of the Korean versions of the COVID-19 Vaccine Hesitancy and Vaccine Conspiracy Beliefs Scales”. I believe the topic is engaging and addresses a relevant issue about the investigation of conspiracy beliefs associated to covid-19 vaccination. I found the manuscript well written and understood the authors’ intent to provide such a tool, since it may have relevant practical implications.

I have minor concerns regarding the current version of the manuscript. I propose a few minor revisions and suggestions that might ameliorate their work.

- I believe that the introductory section of the manuscript, as well as the conclusions, may be improved by integrating recent literature that has highlighted psychosocial variables that may affect the endorsement of conspiracy beliefs related to covid-19 diseases and vaccinations:

1. Douglas,K.M. COVID-19 conspiracy theories. Group Processes Intergroup Relat. 2021, 24, 270–275. https://doi.org/10.1177/1368430220982068.

2.Pellegrini,V.; Giacomantonio, M.; De Cristofaro, V.; Salvati, M.; Brasini, M.; Carlo, E.; Mancini, F.; Leone, L. Is COVID-19 a natural event? COVID-19 pandemic and conspiracy beliefs. Personal. Individ. Differ. 2021, 181, 111011. https://doi.org/10.1016/j.paid.2021.111011.

  1. Giacomantonio, M., Pellegrini, V., De Cristofaro, V., Brasini, M., & Mancini, F. (2022). Expectations about the “Natural Order of Things” and Conspiracy Beliefs about COVID-19. International Journal of Environmental Research and Public Health19(15), 9499. Doi: 10.3390/ijerph19159499

  → The authors added and modify. Please see the red text on page 9-10.

- Please, avoid reporting the GFI as goodness-of-fit indicators since is deprecated.

  → The authors modified Please see the 7page

-Provide more justification on the low fit indexes of KCVH-S Model 1 and on how Authors deal with them.

→ The authors added and modify. Please see the red text on page 7.

- It should be clarified how the convergent and divergent validity of the measure has been tested. Usually, these types of validity are tested through the examination of convergent and divergent associations between the measure of interest and other constructs of theoretical relevance that are potentially related to it. In this paper, this does not appear to have been done. Rather, an analysis of the usefulness of having multiple dimensions in a single measuring instrument seems to have been conducted. Therefore, I would suggest clarifying this aspect and to call the related analyses in an appropriate way.

→ The authors added and modify. Please see the red text on page 10.